# Assessing the Neurotoxicity of a Sub-Optimal Dose of Rotenone in Zebrafish (*Danio rerio*) and the Possible Neuroactive Potential of Valproic Acid, Combination of Levodopa and Carbidopa, and Lactic Acid Bacteria Strains

**DOI:** 10.3390/antiox11102040

**Published:** 2022-10-17

**Authors:** Ovidiu-Dumitru Ilie, Raluca Duta, Ioana-Miruna Balmus, Alexandra Savuca, Adriana Petrovici, Ilinca-Bianca Nita, Lucian-Mihai Antoci, Roxana Jijie, Cosmin-Teodor Mihai, Alin Ciobica, Mircea Nicoara, Roxana Popescu, Romeo Dobrin, Carmen Solcan, Anca Trifan, Carol Stanciu, Bogdan Doroftei

**Affiliations:** 1Department of Biology, Faculty of Biology, “Alexandru Ioan Cuza” University, Carol I Avenue, no 20A, 700505 Iasi, Romania; 2Department of Exact and Natural Sciences, Institute of Interdisciplinary Research, “Alexandru Ioan Cuza” University, Carol I Avenue, no 11, 700506 Iasi, Romania; 3Doctoral School of Biology, Faculty of Biology, “Alexandru Ioan Cuza” University, Carol I Avenue, 20A, 700506 Iasi, Romania; 4Doctoral School of Geosciences, Faculty of Geography-Geology, “Alexandru Ioan Cuza” University, Carol I Avenue, no 20A, 700505 Iasi, Romania; 5Department of Molecular Biology, Histology and Embryology, Faculty of Veterinary Medicine, University of Life Sciences “Ion Ionescu de la Brad”, Mihail Sadoveanu Street, no 3, 700490 Iasi, Romania; 6Faculty of Medicine, University of Medicine and Pharmacy “Grigore T. Popa”, University Street, no 16, 700115 Iasi, Romania; 7Department of Medical Genetics, University of Medicine and Pharmacy “Grigore T. Popa”, University Street, no 16, 700115 Iasi, Romania; 8Research Center on Advanced Materials and Technologies, Department of Exact and Natural Sciences, Institute of Inderdisciplinary Research, “Alexandru Ioan Cuza” University, Carol I Avenue, no 11, 700506 Iasi, Romania; 9Advanced Research and Development Center for Experimental Medicine (CEMEX), University of Medicine and Pharmacy “Grigore T. Popa”, University Street, no 16, 700115 Iasi, Romania; 10Department of Medical Genetics, “Saint Mary” Emergency Children’s Hospital, Vasile Lupu Street, no 62, 700309 Iasi, Romania; 11Department of Psychiatry, University of Medicine and Pharmacy “Grigore T. Popa”, University Street, no 16, 700115 Iasi, Romania; 12Department of Gastroenterology, University of Medicine and Pharmacy “Grigore T. Popa”, University Street, no 16, 700115 Iasi, Romania; 13Institute of Gastroenterology and Hepatology, “St. Spiridon” Emergency Hospital, Independence Avenue, no 1, 700111 Iasi, Romania

**Keywords:** Parkinson’s disease, zebrafish, rotenone, valproic acid, levodopa, carbidopa, probiotics

## Abstract

Parkinson’s disease (PD) is an enigmatic neurodegenerative disorder that is currently the subject of extensive research approaches aiming at deepening the understanding of its etiopathophysiology. Recent data suggest that distinct compounds used either as anticonvulsants or agents usually used as dopaminergic agonists or supplements consisting of live active lactic acid bacteria strains might alleviate and improve PD-related phenotypes. This is why we aimed to elucidate how the administration of rotenone (ROT) disrupts homeostasis and the possible neuroactive potential of valproic acid (VPA), antiparkinsonian agents (levodopa and carbidopa – LEV+CARB), and a mixture of six *Lactobacillus* and three *Bifidobacterium* species (PROBIO) might re-establish the optimal internal parameters. ROT causes significant changes in the central nervous system (CNS), notably reduced neurogenesis and angiogenesis, by triggering apoptosis, reflected by the increased expression of *PARKIN* and *PINK1* gene(s), low brain dopamine (DA) levels, and as opposed to *LRRK2* and *SNCA* compared with healthy zebrafish. VPA, LEV/CARB, and PROBIO sustain neurogenesis and angiogenesis, manifesting a neuroprotective role in diminishing the effect of ROT in zebrafish. Interestingly, none of the tested compounds influenced oxidative stress (OS), as reflected by the level of malondialdehyde (MDA) level and superoxide dismutase (SOD) enzymatic activity revealed in non-ROT-exposed zebrafish. Overall, the selected concentrations were enough to trigger particular behavioral patterns as reflected by our parameters of interest (swimming distance (mm), velocity (mm/s), and freezing episodes (s)), but sequential testing is mandatory to decipher whether they exert an inhibitory role following ROT exposure. In this way, we further offer data into how ROT may trigger a PD-related phenotype and the possible beneficial role of VPA, LEV+CARB, and PROBIO in re-establishing homeostasis in *Danio rerio*.

## 1. Introduction

PD defines an irreversible and unidirectional degeneration of dopaminergic neurons from the substantia nigra (SN), a process restricted to the ventrolateral segment in the prodromal stages. The consequences include DA biosynthesis inhibition followed by specific α-synuclein (αS)-positive Lewy body (LB) aggregates. Bradykinesia, essential tremor, and stiffness are typical signs for a proper differential diagnosis, while the impairment of the postural balance occurs later in the patient’s life [1,2]. However, PD-diagnosed patients might also display non-motor symptoms, including sleep impairments and autonomic, sensory, and gastrointestinal deficiencies [1].

Studies aiming to establish the actual number of individuals suffering from PD are lacking. However, based on the estimations, the prevalence (6.1 million) and mortality (3.2 million) are comparable to those of Alzheimer’s disease (AD), and PD may surpass AD in the next couple of decades. Prevalence in Europe ranges from 65.5 to 12,500, while incidence annually oscillates from 5 to 346 per 100,000 inhabitants [3], strongly related to age, sex, and geographical location, particularly in men in their fifties [4].

Based on the classification criteria (stage I–V) introduced by Hoehn and Yahr in 1967 [5], and since there is no typical age for the onset, PD could be juvenile (<20), early-onset (20–50), or late-onset (>50) [6]. PD possesses a multifactorial nature [7,8] with circumstances where patients display specific symptoms as a side effect of the antipsychotics administered, a syndrome known as drug-induced parkinsonism (DIP) [9]. The actual nomenclature describes several subtypes of PD, including idiopathic PD and LB dementia [10].

Unfortunately, the exact origin of PD remains elusive despite the best efforts. Apiece from the typical symptomatic medication, there are also surgery and conventional approaches that target the alleviation of locomotor impairments. Currently, there is no other non-invasive management strategy to diminish or even treat PD. The environmental hypothesis has been postulated and supported by the efforts of scientists to generate specific models [11]. A correlation following the exposure to natural-derived compounds with a toxic potential [12] and a PD phenotype in zebrafish (*Danio rerio*) [13], humans [14], and rodents [15] was achievable.

ROT crosses the blood–brain barrier (BBB) and accumulates in cellular organelles, particularly mitochondria [16,17]. From this point, it causes neuronal toxicity [18] and a decline in adenosine triphosphate (ATP) production. As the homeostasis is gradually disturbed, this physiological response is followed by an exacerbation in reactive oxygen species (ROS) generation via the inhibition of the mitochondrial electron transport chain (ETC) [19], enabling the activation of microglia, neuroinflammation [20], and abnormal aggregation of αS responsible for LB pathology [21].

Based on the existing literature that describes an inhibitory role of VPA in ROT-exposed rats, we hypothesized that VPA might counter the antagonistic effects of ROT in zebrafish as well [22,23]. The antiparkinsonian combination of LEV/CARB is the last line of defense in PD-diagnosed patients. It downregulates cortisol levels in zebrafish via the hypothalamic–pituitary–adrenal axis (HPA) [24]. PROBIO is another powerful category of supplements due to its non-invasive potential, enhancing the host’s eubiosis [25]. Compared with the aforementioned, PROBIO cannot cross the BBB, but the responses initiated influence the brain’s reactivity through the intracellular cascades emitted. According to the evidence in the field, exogenous supplementation with PROBIO enhances the host’s gut microflora integrity by preventing dysbiosis. In this context, microorganisms that reside within the gut negate the pathogens’ endotoxins, which otherwise cross the BBB and trigger neuroinflammation. Thus, a regime might impact the host composition through competition, antagonism, and cross-feeding [26,27]. Therefore, the present study aims to offer systematic insights by assessing VPA, LEV/CARB, and PROBIO effects in a zebrafish (*Danio rerio*) model of PD chronically exposed to sub-optimal dose concentration of ROT for 32 days.

## 2. Materials and Methods

### 2.1. Ethical Approval

Animals were maintained and treated in accordance with the EU Commission Recommendation (2007), Directive 2010/63/EU of the European Parliament, and the Council of 22 September 2010 guidelines for accommodation, care, and protection of animals used for experimental and other scientific purposes. The current protocol received approval from the Ethics Committee of the Faculty of Biology, “Alexandru Ioan Cuza” University, Iasi, with the registration number 3936/26/11/2021. All procedures performed aimed to use a limited number of zebrafish and respect current legal guidelines and regulations and Animal Research: Reporting of In Vivo Experiments (ARRIVE) guidelines [28].

### 2.2. Animal Husbandry

One hundred twenty adult (6–8 months) wild-type AB (WT, genetic line) zebrafish (*Danio rerio*) purchased from an authorized local breeder were housed for 14 days in a 90 L recirculating dechlorinated water aquarium equipped with a thermometer and water recirculation system. Subsequently, we divided zebrafish into eight equal groups (*n* = 15) by placing them into new 10 L aquariums for an additional week. We applied this strategy, seeking to familiarize the zebrafish with the stress of being caught and transferred into tanks with the novel configuration. The standard laboratory conditions were: temperature of 26 ± 2 °C; pH 7.5; 14 h light/10 h night cycle [29]. Following the water change in each experimental tank daily, the feeding routine involved TetraMin Flakes twice per day.

### 2.3. Administration of Compounds and Supplements

ROT (#R700580) (5 g) and VPA (#SLBC9758V) (100 g) were acquired from Toronto Research Chemicals, North York, Canada, and Sigma-Aldrich, Saint Louis, MO, USA, whereas LEV/CARB (250 mg + 25 mg—10 pills per blister × 3) and PROBIO (3 g × 28 envelopes) were acquired from a local pharmacy. Products’ commercial label remains anonymous to avoid any conflicts of interest for these products. PROBIO contains the following bacteria strains: *Lactobacillus casei* W56, *Lactobacillus acidophilus* W22, *Lactobacillus paracasei* W20, *Lactobacillus salivarius* W24, *Lactobacillus lactis* W19, *Lactobacillus plantarum* W62, *Bifidobacterium lactis* W51, *Bifidobacterium lactis* W52, and *Bifidobacterium bifidum* W23. ROT and VPA had a solid state of white-like powder that was dissolved in distilled water until we reached 2.5 µg/L and 0.5 mg/mL, while LEV/CARB and PROBIO were dissolved directly into 100 mL distilled water as a unique dose per day using a rated balloon. LEV/CARB and PROBIO were administered half an hour before routine feeding to ensure proper ingestion [30].

As already demonstrated by our group [31] and concomitantly by another team [32], administration of 2 µg/L ROT for 21 to 28 days triggers a mild locomotor impairment in zebrafish. On the other hand, 5 µg/L ROT [33,34,35,36] exposure for 28 [33,35] to 30 days [34,36] led to visible locomotor dysfunctionalities. Due to excessive mortality approximately ten days after inception that occurred in a preliminary study conducted by us following the administration of 5 µg/L ROT (data not shown), we decided to halve the dose.

In the same preliminary experiment we also tested four concentrations of VPA (0.5 mg/mL, 2 mg/mL, 5 mg/mL, and 10 mg/mL). In zebrafish receiving 5 mg/mL and 10 mg/mL, we noted high mortality in the first 6–12 h post-administration, while those exposed to 2 mg/mL were immobile upon touching (data not shown). In this way, we concluded that 0.5 mg/mL might be the optimum dose since a study in which we applied this approach was already published [37].

In what concerns the associated dose of LEV/CARB, we used as informatic support the study of Idalencio et al. [24], where the authors emphasize the role of LEV/CARB in stress response in zebrafish in contrast with non-stressed fish, suggesting that DA was related to the balance between high and low cortisol levels and that norepinephrine (NE) decreased this response.

There are no restrictions regarding PROBIO dose and ratio of bacteria; they are customized depending on the study design, product strains, and colony-forming units (CFU) rather than sex and species. For example, Valcarce et al. [30,38] administered a 1:1 ratio of *Lactobacillus rhamnosus* CECT8361 and *Bifidobacterium longum* CECT7347 in both *Danio rerio* and human patients to assess how short- and long-term administration of these two PROBIO strains improve sperm quality. In another study, Valcarce et al. [39] used the same combination to alter swimming patterns and speed in zebrafish by downregulating anxiety-related behavior. In one study of ours, *Bifidobacterium longum* BB536 and *Lactobacillus rhamnosus* HN001 were administered to 2 µg/L ROT-exposed zebrafish to investigate their possible role in sociability and locomotion. We concluded that the selected dose is insufficient to trigger PD, only initiating mild symptoms [31].

The studied groups were as follows: (a) CONTROL, (b) VPA, (c) LEV/CARB, (d) PROBIO, (e) ROT, (f) ROT+VPA, (g) ROT+LEV/CARB, (h) ROT+PROBIO. At the end of the analyzed period, zebrafish were euthanized by immersion in ice-cold water at 2–4 °C for 10 min until the disappearance of the operculum movements [40].

### 2.4. Locomotor Testing

Locomotor activity testing was performed using a 10 L trapezoidal tank filled with 6 L dechlorinated tap water. Parameters of interest were: the total distance (mm), velocity (mm/s), and freezing duration (s). The conformations were analyzed using the Track3D module of EthoVisionXT 14 video tracking software (Noldus Information Technology, Wageningen, the Netherlands). The trial lasted 4 min with 30 s of accommodation for each subject.

### 2.5. Oxidative Stress Marker Measurements

Tissue homogenates for the evaluation of oxidative biomarkers were acquired by following the protocol described in our work [31]. We assessed the enzymatic activity of SOD (EC 1.15.1.1) using a determination kit (19160-1KT-F) from Sigma-Aldrich, Darmstadt, Germany, while that of the lipid peroxidation marker MDA was assessed using a predetermined protocol [41]. We set the Specord 210 Plus (Analytik Jena, Germany) at distinct colorimetric wavelengths (320 nm for SOD and 532 nm for MDA). Results for OS are reported as U SOD/mg protein, and results for MDA as % of CONTROL.

### 2.6. Real-Time Polymerase Chain Reaction

We obtained the total RNA from the brain using peqGOLD TriFast (Peqlab Biotechnologie, Erlangen, Germany), while the reverse transcription was performed with GoTaq 1-Step RT-qPCR (Madison, WI, USA) according to the supplier’s instructions and using 4 µL of RNA to synthesize the cDNA. The amplification reactions were performed on AriaMx Real-Time PCR System (Agilent, Santa Clara, CA, USA) and involved five PD-related genes. Forward and reverse sequences from Integrated DNA, San Diego, CA, USA, for PINK1, PARKIN, LRRK2, alpha-SNCA, and ACTIN are as follows: *PINK1* (NM_001008628.1) (2088 bp), f: 5′-GGCAATGAAGATGATGTGGAAC-3′, r: 5′-TTGTGGGCATGAAGGAACTAAC-3′; *PARKIN* (NM_001017635.1) (1465 bp), f: 5′-GAGGAGTTTCACGAGGGTCC-3′, r: 5′-TGAGTGGTTTTGGTGATGGTC-3′; *LRRK2* (NM_001201456.2) (9170 bp), f: 5′-ACTCGGATTAAGTTCCACCAGA-3′, r: 5′-CAGTGAGGGTTGATGGTCTGTA-3′; *alpha-SNCA* (NM_001017567.2) (1294 bp), f: 5′-ATGCACTGAAGAAGGGATTCTC-3′, r: 5′-AGATTTGCCTGGTCAGTTGTTT-3′; and *ACTIN* (NM_181601.5) (1843 bp), f: 5′-GGCATCACACCTTCTACAATGA-3, r: 5′-TACGACCAGAAGCGTACAGAGA-3′.

We also applied a melting curve to increase the specificity and exclude the possible derived products associated with primer dimers following amplification. The amplification protocol was: RT: 20:00 min at 38 °C (1 cycle), hot-start: 10:00 min at 95 °C (1 cycle), amplification: 00:10s at 95 °C, 00:30 s at x °C (depending on the gene), 00:30 s at 72 °C (40 cycles), melting curve: 00:30 s at x °C (depending on the gene) × 2 and 00:30 s at 95 °C (1 cycle); *ACTIN*, *PINK1,* and *SNCA* at 55 °C; *LRRK2* at 57.5 °C; and *PARKIN* at 58.5 °C. The relative expression following RT-PCR was determined using the formula 2^−∆∆Ct^, the final results being normalized to *ACTIN*.

### 2.7. Dopamine Measurements

Brain DA was assessed through enzyme-linked immunosorbent assay (ELISA) using the Fish Dopamine ELISA kit (antibodies, Aachen, Germany) according to the supplier’s instructions. The readings were taken at 450 nm wavelength using Tecan Sunrise (Tecan, Crailsheim, Germany). Calculations for DA using standards were performed using a polynomial regression with the formula: y =a + bx + c/x^2.

### 2.8. Histological and Immunohistological Analysis

Samples from distinct organs were subjected to the standard fixation in Bouin solution for 24 h, and approximately 0.5 cm thick slices were subsequently dehydrated using a decreasing concentration of ethanol solution and then clarified in xylene and embedded in paraffin. After being cut with the microtome, 10 microscope slides from each paraffin block were selected, specifically stained, and observed under an Olympus CX41 light microscope (Olympus Europa SE & Co, Wendenstraße, Hamburg, Germany). They were initially stained with hematoxylin–eosin (HE) and then IHC stained using 5 antibodies: glial fibrillary acidic protein (GFAP), S100 calcium-binding protein B (S100B), proliferating cell nuclear antigen (PCNA), tumor protein p53, and cytochrome C oxidase subunit 4I1 (cox4i1). After the sections were deparaffinized in xylene, hydrated in ethanol, and microwaved for 10 min at 95 °C in 10 mmol citrate acid buffer pH 6, they were left to cool down for 20 min and then washed twice in phosphate-buffered saline (PBS) for 5 min. Slices were treated with 3% hydrogen peroxide and rinsed with PBS, after which they were incubated overnight at 4 °C in a humid atmosphere with primary antibodies in dilutions of 1:1000 GFAP and S100b; 1:250 for PCNA, p53, and Cox4i1. The next day, slides were washed 3 times in PBS for 5 min and incubated with secondary antibodies. Goat anti-rabbit IgG secondary antibody was used for GFAP, S100b, p53, and PCNA. Microscope slides were developed in 3,3′-diaminobenzidine (DAB) and finally counterstained with hematoxylin.

### 2.9. Statistical Analyses

The normality and distribution of behavioral data were determined by the Shapiro–Wilk test using Graph Pad Prism software (v 9.1.0.221, San Diego, CA, USA). Subsequently, comparisons between groups were performed with unidirectional ANOVA followed by post hoc parametric tests such as Tukey HSD. Data are expressed as mean with standard error of the mean (SEM). A *p* < 0.05 was considered statistically significant, each analysis being conducted in duplicate using *n* = 5 individuals, except RT-PCR where we used only *n* = 3 individuals. We applied RT negative controls for each targeted gene to ensure no DNA contamination.

## 3. Results

### 3.1. ROT Did Not Significantly Influence the Locomotor Performances in Exposed Zebrafish

To evaluate the toxicological effect of ROT after 32 days of administration and the possible neuroactive potential of VPA, LEV/CARB, and PROBIO, we first performed the three-dimensional (3D) locomotor activity test. In Figure 1, several spatial conformations of the reconstructed 3D swimming routes can be observed. The swimming distance (mm), velocity (mm/s), and freezing episodes (s) are further detailed in Figure 2, Figure 3 and Figure 4.

As expected in the case of the four non-ROT-exposed groups (a–d), zebrafish retain their exploratory behavior throughout the entire analyzed period, with slight phenotypical changes in (b) VPA and (c) LEV/CARB defined as “reaching the surface of the water” indicating the possible neuroactive potential of these two agents, whereas in the (d) PROBIO group a behavioral pattern similar to the (a) CONTROL can be seen. On the other hand, the groups given ROT (e–h) manifest an affinity towards the bottom of the aquarium as observed in (e) ROT and (g) ROT+LEV/CARB. Moreover, the swimming patterns in (f) ROT+VPA and (h) ROT+PROBIO are antithetical in contrast with (e) ROT and (g) ROT+LEV/CARB.

Regarding the swimming distance (mm) parameter recorded, significant differences emerge between the (a) CONTROL and several experimental groups: compared with (e) ROT, *p* = 0.002/*p* < 0.0001, and (h) ROT+PROBIO, *p* = 0.004/*p* = 0.000, according to both cameras, whereas in (d) PROBIO, *p* = 0.002; (f) ROT+VPA, *p* = 0.004; and (g) ROT+LEV/CARB, *p* = 0.000 based on the trials registered only by the lateral camera. Other circumstances where we noted differences were in (b) VPA compared with (e) ROT, *p* = 0.009/*p* = 0.038, also according to both cameras, and (h) ROT+PROBIO, *p* = 0.014, by the top camera. Additionally, the group (c) LEV/CARB exhibited differences with (e) ROT, *p* = 0.001; (g) ROT+LEV/CARB, *p* = 0.022; and (h) ROT+PROBIO, *p* = 0.025 (Figure 2).

Intriguing behavioral patterns further arise following the centralization of data attributed to the velocity (mm/s) parameter in (a) CONTROL by comparison with both ROT-exposed and non-exposed groups. More specifically, the only group where we did not note a significant difference was by comparison with (b) VPA (*p* > 0.05) and with a particular case in (c) LEV/CARB, *p* = 0.024 per lateral camera. Continuing with this concept, we still found major differences between the healthy zebrafish and (d) PROBIO, *p* = 0.004/*p* = 0.004; (e) ROT, *p* = 0.000/*p* = 0.000; (f) ROT+VPA, *p* = 0.008/*p* = 0.007; (g) ROT+LEV/CARB, *p* = 0.019/*p* = 0.005; and (h) ROT+PROBIO, *p* = 0.003/*p* = 0.000, based on top and lateral camera recordings (Figure 3).

Complementary to the observations made in the light of the above two parameters, in (a) CONTROL, multiple episodes of inactivity (s) were observed by reference with (c) LEV/CARB, *p* = 0.000/*p* = 0.010, and (d) PROBIO, *p* = 0.031/*p* = 0.042 via the top and side view and between (e) ROT, *p* = 0.007, and (g) ROT+LEV/CARB, *p* = 0.036, by the lateral camera. Two sole cases in which we observed differences were in (b) VPA in contrast with (c) LEV/CARB, *p* = 0.013, by the vertical camera, and (e) ROT with (f) ROT+VPA, *p* = 0.045, by the lateral camera (Figure 4).

### 3.2. ROT Causes a Reduction in Neurogenesis and Triggers Apoptosis in Exposed Zebrafish

At the level of the optic tectum in the (a) CONTROL group, a moderate marking based on the immunohistochemical (IHC) markers used was observed. PCNA marks two small areas of neuronal stem cells (NSCs) and neuroblasts, respectively.

In (b) VPA, areas of neurogenesis in the mesencephalon and angiogenesis in the periventricular gray area can be observed. PCNA, S100b, GFAP, and cox4i1 markers exhibit an intense expression in torus longitudinalis, torus semicircularis (gray periventricular area), and basal tegmentum. The immunoreactivity of S100b protein was detected particularly in the mesencephalic optic tectum, marking nerve fibers, particularly profiles rather than cells. In fact, immunoreactivity was concentrated in the fiber profiles that cross the entire optical tectum perpendicular to the outer side. Moreover, the immunoreactivity of protein S100b was further located in the medial and lateral areas of the cerebral valvula. The internal area of the optic tectum located in the tectal ventricle was also lined with cells positive for S100b and GFAP proteins that exhibit morphological characteristics of ependymal cells and subependymal cells. Ependymal cells are large, and of round shape, while subependymal cells and glial cells have long radial processes that pass through the optic tectum and reach the pial surface. The dorsal and lateral parts of the torus longitudinalis were covered by ependymal cells that have the S100b protein, and the nerve fibers that form the commissure in the ventral part of the torus longitudinalis were also positive for the S100b protein.

Group (c) LEV/CARB showed islands of PCNA cells compared to the other groups, newly formed capillaries in the diencephalon and mesencephalon. S100b and GFAP markers were positively labeled in a larger number of cells, while cox4i1 and p53 had a moderate label.

In (d) PROBIO group, a positive marking was registered, especially for PCNA, S100b, and GFAP and moderately for p53 and cox4i1.

On the other hand, in (e) ROT, reduction to absence was noted for the PCNA, GFAP, S100, and moderate markers were noted for p53 and cox4i1.

In (f) ROT+VPA, (g) ROT+LEV/CARB, and (h) ROT+PROBIO, there was an increase in all IHC markers, and in (h) ROT+PROBIO PCNA marked a small number of cells.

ROT caused a reduction in PCNA labeling, which may suggest a decrease in neurogenesis and, implicitly, an increase in neuronal dysfunction via the reduction in GFAP and S100b labeling. This might be explained by the expression of p53 and cox4i1 labeling, which further indicates apoptosis and mitochondrial dysfunction.

However, in groups (b) VPA, (c) LEV/CARB, and (d) PROBIO, there is a positive marking for neurogenesis (PCNA), apoptosis (p53), and the presence of mature, active radial neurons and glial cells (GFAP and S100b). Cox4i1 has positivity in these groups, which denotes intense mitochondrial activity.

In the cerebellum, radial glial (RG) cells and neurons from the gray matter in the molecular layer, Purkinje and granular, were highlighted, marked with GFAP, S100b, p53, and cox4i1 of close intensity with the optical tectum from each experiment. The pattern of distribution of the S100b protein in the cerebellum differs from that of other segments of the CNS because it has been located mainly in neurons, more frequently than in glial cells. In the body of the cerebellum, the S100b protein marked the small neurons in the superficial, molecular layer. In addition, the neurons that make up the cerebellar body and deep nuclei were also immunoreactive for S100b. Purkinje neurons located in the basal area showed a strong reaction to the S100b protein in both the perikaryon and the dendritic tree (Figure 5).

### 3.3. VPA, LEV/CARB, and PROBIO Do Not Attenuate Oxidative Stress in ROT-Exposed Zebrafish

The evaluation of oxidative biomarkers revealed a statistically significant difference in the level of MDA between (b) VPA and (c) LEV/CARB (*p* = 0.023) and compared to (d) PROBIO (*p* = 0.009). There are also changes in the enzymatic activity of SOD when comparing the same groups (*p* = 0.025). The SOD activity is lower in (e) ROT compared to (c) LEV/CARB (*p* = 0.034), in relation to the MDA level in the group (e) ROT. Contrary to our expectations, there is no difference between the (a) CONTROL group and the experimental groups exposed to ROT. Even though the activity of SOD in (f) ROT+VPA is similar to that of (e) ROT, there is no noticeable difference when comparing the results between ROT-exposed groups (*p* > 0.05). Slight instabilities in the level of MDA in (f) ROT+VPA, (g) ROT+LEV/CARB, and (h) ROT+PROBIO analogous to (a) CONTROL and (h) ROT+PROBIO in SOD are observable (Figure 6).

### 3.4. LEV/CARB Induce Changes Only in the Expression of PARKIN, Whereas ROT Alone Induces Changes in the Expression PINK1 in Exposed Zebrafish

By analyzing the relative expression of four PD-related genes in ROT-exposed zebrafish, it was found that there was no statistically significant difference in *LRRK2* and *SNCA* (*p* > 0.05). There are statistically significant differences in *PARKIN* due to overexpression in (g) ROT+LEV/CARB and each experimental group. Therefore, we noted in (a) CONTROL vs. (g) ROT+LEV/CARB, *p* = 0.000; (b) VPA vs. (g) ROT+LEV/CARB, *p* = 0.000; (c) LEV/CARB vs. (g) ROT+LEV/CARB, *p* = 0.000; (d) PROBIO vs. (g) ROT+LEV/CARB, *p* = 0.000; (e) ROT vs. (g) ROT+LEV/CARB, *p* = 0.001; (f) ROT+VPA vs. (g) ROT+LEV/CARB, *p* = 0.002; and (g) ROT+LEV/CARB vs. (h) ROT+PROBIO—*p* = 0.007. Finally, another exacerbated expression of *PINK1* in (e) ROT promoted three situations in which we observed a statistically significant difference: (d) PROBIO vs. (e) ROT, *p* = 0.007; (e) ROT vs. (g) ROT+LEV/CARB, *p* = 0.010; and (e) ROT vs. (h) ROT+PROBIO, *p* = 0.008 (Figure 7).

### 3.5. VPA, LEV/CARB, and PROBIO Do Not Influence DA Levels in ROT-Exposed Zebrafish

The DA level reached a peak in the group (a) CONTROL, surpassing all experimental groups except (b) VPA, where there is no significance. Precisely, there are statistical differences in the following cases: (c) LEV/CARB, *p* = 0.000; (d) PROBIO, *p* = 0.000; (e) ROT, *p* = 0.001; (f) ROT+VPA, *p* = 0.002; (g) ROT+LEV/CARB, *p* <0.0001; and (h) ROT+PROBIO, *p* = 0.000. Also, in group (b) VPA, there were three situations in which the DA level is elevated compared to (c) LEV/CARB, *p* = 0.024; (g) ROT+LEV/CARB, *p* = 0.009; and (h) ROT+PROBIO, *p* = 0.048 (Figure 8).

## 4. Discussion

In this manuscript, we aimed to assess the potential neuroactive ability of VPA, the combination of LEV/CARB, and PROBIO to counter the neurotoxic effect of ROT in zebrafish after 32 days of chronic exposure to a sub-optimal dose concentration. For this, we preliminarily tested several doses and concluded that 2.5 µg/L ROT and 0.5 mg/mL VPA are the optimum dose concentrations to prevent excessive mortality despite the existing literature in which the authors administered 5 µg/L ROT for 28 [33,35] to 30 days [34,36]. In our case, however, it has been proved that 2.5 µg/L ROT did not significantly influence any of the assessed parameters. The animals receiving this dose exhibited normal behavior above the average compared to the rest of the experimental groups since they registered the longest swimming distance with the highest speed and the fewest freezing episodes. Our results are congruent with those of our previous study [31] and the study of Wang et al. [32] following the administration of 2 µg/L ROT for 21/28 days.

However, from this point, three case scenarios that may explain this situation can be derived: zebrafish begin to metabolize the administered ROT, the route of administration does not ensure the appropriate ingestion of the compound, or the exposure period is not sufficient at the selected dose concentration. Moreover, there are other variables concerning stress-related behavioral analyses. We hypothesized that daily testing could accelerate the metabolic rate, which may further boost and potentiate ROT’s effect, reflected by the abnormal generation of free radicals. On the other hand, testing at specific intervals might not exert a powerful impact on physiology. Unfortunately, most of the current data concerning locomotor impairments were reported per week [33,35] or not specified [34,36] and did not concern ROT administration or substance renewal.

The main neurogenic niches studied in adult zebrafish are the telencephalon, optic tectum, and cerebellum. From these structures, the telencephalon remains the most examined region of the brain since it shares anatomical structures and homology with mammals [42,43,44,45]. As other studies indicate, telencephalon and diversity of neuronal/progenitor cells support firm neurogenic activity observed in the various telencephalic subdomains of the zebrafish brain [46,47,48].

Most neural stem cells (NSCs) under homeostatic conditions are considered quiescent type I radial glia (RGCs), a pattern notable in the (a) CONTROL group. However, a small proportion of NSCs proliferates and express proliferation markers such as PCNA. These so-called type II RGCs can give rise to employed neuronal progenitors that correspond to neuroblasts (type III cells) [49,50]. When the telencephalon is damaged and subjected to aggression, several NSCs are activated, enter the cell cycle, and begin to express proliferation markers [50], a situation observed in each experimental group except (e) ROT. NSCs generate an increased number of new neurons compared to homeostatic conditions [51], newborn neuronal precursors migrating from the ventricular layer to the site of the lesion to replace the lost neurons, and regenerative neurogenesis triggered by inflammatory signals [52,53,54]. Additionally, in all experimental groups except (e) ROT, an intense expression of ventricular GFAP and S100b of the telencephalon and mesencephalon was observed, which are densely populated by the cell bodies of radial glia cells (RGPs). VPA attenuates tissue and neuronal damage, improves functional restoration, and stimulates neurogenesis and operational integration. This stimulation of neurogenesis was seen in (b) VPA and (f) ROT+VPA. In this experiment, we suggested that the effects of exogenous LEV observed in (g) ROT+LEV/CARB were due to its absorption into the neuronal or glial system in the cytoplasm of cells. Glial cells of the subventricular area play an essential role in the neurogenesis of adult zebrafish [55], which explains the presence of GFAP and S100b expression in the subventricular area in fish in all experimental groups except (e) ROT. The S100 protein could be maintained in this function in zebrafish for life [56,57]. Interestingly, angiogenesis was observed in this experiment on three occasions: (b) VPA, (c) LEV/CARB, and (g) ROT+LEV/CARB.

Although the overproduction of ROS is well known to be a significant trigger of various sequential consequences which culminate in dopaminergic neuron destruction, mitochondrial dysfunction may affect αS in a feedback loop, regulated by the expression of the SNCA coding gene [58]. The enzymatic activity of SOD and the MDA level in the CONTROL group were non-significantly higher than those of ROT (*p* > 0.05) and lower than those receiving *Bifidobacterium longum* BB536 and *Lactobacillus rhamnosus* HN001 alone or in combination with ROT in the case of MDA (*p* < 0.05) [31]. While there were slight differences in the level of MDA in contrast with the activity of SOD, these levels reached a peak in the (c) LEV/CARB and (d) PROBIO groups. Interestingly, there were no significant differences between the (a) CONTROL and the experimental groups, particularly in ROT-exposed zebrafish. Although the level of MDA is lower in contrast with the four groups exposed to ROT, there are no significant changes, analogous to the enzymatic activity of SOD, which objectively is higher for the same groups, but non-significantly (*p* > 0.05). These data disagree with our previous study [31], but this is the second study that assesses oxidative biomarkers at low dose concentrations using the whole fish, not only the brain or the intestine [34,36]. While the activity of SOD is higher in healthy adult zebrafish, ROT-treated zebrafish had an elevated level of MDA in the brain compared to healthy individuals and in the intestine in some circumstances. The authors partially corrected this imbalance by administrating caprylic acid, mitoquinone, and oleandrin following the administration of 5 µg/L ROT [34,36]. One possible explanation that should be tested for the lack of significance is that VPA, LEV/CARB, and PROBIO combined with ROT exert an antagonist phenomenon, canceling each other’s effects. The expression levels of *PINK1* and *PARKIN* genes were correlated to mitophagy, and recent studies suggest that mutations in the *LRRK2* and *SNCA* also contribute to mitochondrial dysfunction [34]. Our results further strengthen the observations made by Ünal et al. [34] and contradict those of Wang et al. [32]. While ROT (e–h) causes an increase in the expression of *LRRK2* compared with non-ROT (a–d), it does not inhibit the expression of *SNCA*, a trend similar to that of *LRRK2*. Moreover, ROT increases the expression of *PINK1*, and of *PARKIN*, possibly attributed to the neuroactive potential of LEV/CARB. The results of Ünal et al. [34] and of Wang et al. [32] are confirmed to some extent since they argue that ROT decreases the expression of *PINK1* and of *PARKIN* but in particular cases.

DA is particularly unstable and can self-oxidize to form ROS after degradation by monoamine oxidase B (MAO-B) [59,60]. Brain DA level in the group (a) CONTROL was high, supported by a statistically significant difference compared to the other groups, in parallel with the three situations in the group (b) VPA. *Centella asiatica* [33] and mitoquinone [34] increase DA levels in ROT-exposed zebrafish suffering depletion caused by ROT exposure [35]. One of the most eloquent examples of the effect of ROT on the level of DA is provided by the studies of Alam and Schmidt [61] and Biehlmaier et al. [62]. Intraperitoneal (ip) injection with 1.5 mg/kg, 2 mg/kg/day, and 2.5 mg/kg between ten days and two months induce degeneration of dopaminergic neurons [63] in the posterior part of the striatum, the prefrontal cortex (PFC), and SN. A deletion in the *Ndufs4* gene impacts the complex I activity in mesencephalic neurons but does not affect survival [64].

### Limitations of the Study

We did not follow the quantification of protein expression through ELISA or Western blotting.

## 5. Conclusions

ROT is a potent agent with a toxicological profile capable of causing PD-like symptoms in a time- and dose-dependent manner since a 32-day treatment was long enough to reduce neuro- and angiogenesis and trigger apoptosis in zebrafish. ROT did not significantly influence the behavioral parameters analyzed or the oxidative biomarkers in combination with VPA, LEV/CARB, and PROBIO. PROBIO alone induced a particular phenotype reflected in all three parameters assessed. ROT however induced an increase in two out of four PD-related genes, a decrease in DA brain level, and changes in the following immunohistological examination compared to (a) CONTROL and the other three non-ROT-exposed groups. Even though the administered VPA, LEV/CARB, and PROBIO confer slight neuroprotection against the antagonistic effects of ROT after 32 days of administration through neuro- and angiogenesis and apoptosis, the last two perturb the antioxidant balance systems, probably due to the hyperlocomotion pattern observed in this group.

## Figures and Tables

**Figure 1 antioxidants-11-02040-f001:**
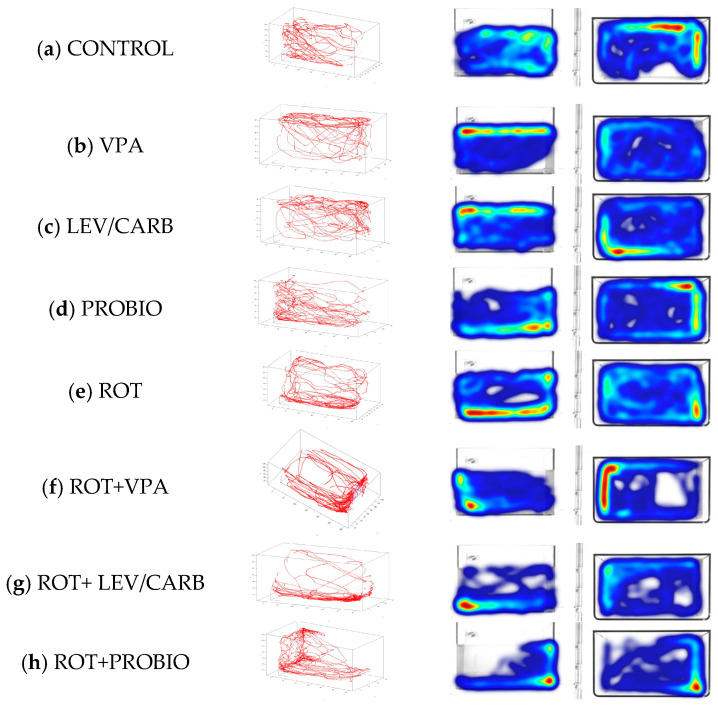
The 3D swim trace results for *Danio rerio* (*n* = 5) in (**a**) CONTROL and the groups treated with (**b**) VPA—0.5 mg/mL, (**c**) LEV/CARB—250 mg + 25 mg, (**d**) PROBIO—3 g, (**e**) ROT—2.5 µg/L, (**f**) ROT+VPA—2.5 µg/L + 0.5 mg/mL, (**g**) ROT+LEV/CARB—2.5 µg/L + 250 mg LEV + 25 mg CARB, and (**h**) ROT + PROBIO—2.5 µg/L + 3 g. An automated integration of traces using Track3D software results in 3D swim tracks reflected by the red color, the right panel indicating the top (X, Y) and side (Y, Z) views of *Danio rerio* in the 240 s test.

**Figure 2 antioxidants-11-02040-f002:**
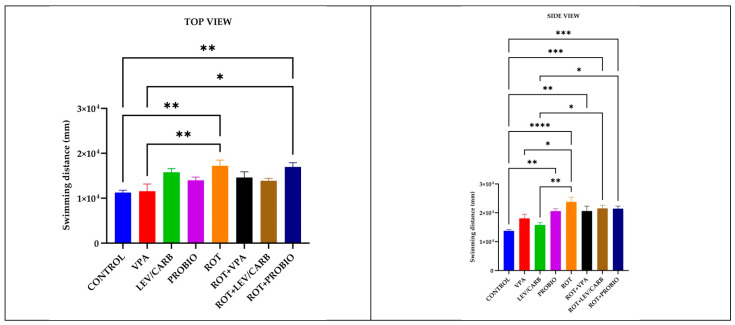
Swimming distance (mm) patterns in studied *Danio rerio* (*n* = 5) groups (*—*p* < 0.05; **—*p* < 0.005; ***—*p* < 0.0005; ****—*p* < 0.0001; values expressed as mean with SEM followed by Tukey test).

**Figure 3 antioxidants-11-02040-f003:**
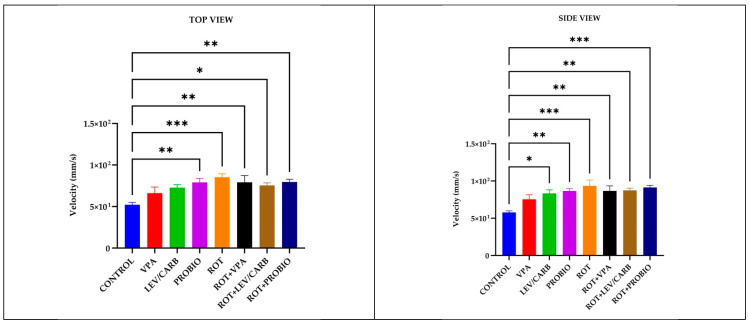
Velocity (mm/s) patterns in studied *Danio rerio* (*n* = 5) groups (*—*p* < 0.05; **—*p* < 0.005; ***—*p* < 0.0005; values expressed as mean with SEM followed by Tukey test).

**Figure 4 antioxidants-11-02040-f004:**
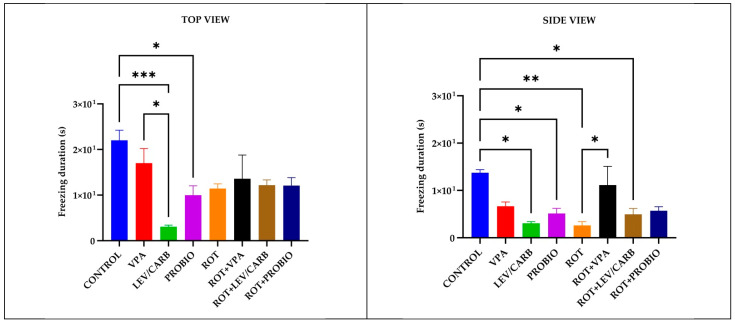
Episodes of inactivity (s) patterns in studied *Danio rerio* (*n* = 5) groups (*—*p* < 0.05; **—*p* < 0.005; ***—*p* < 0.0005; values expressed as mean with SEM followed by Tukey test).

**Figure 5 antioxidants-11-02040-f005:**
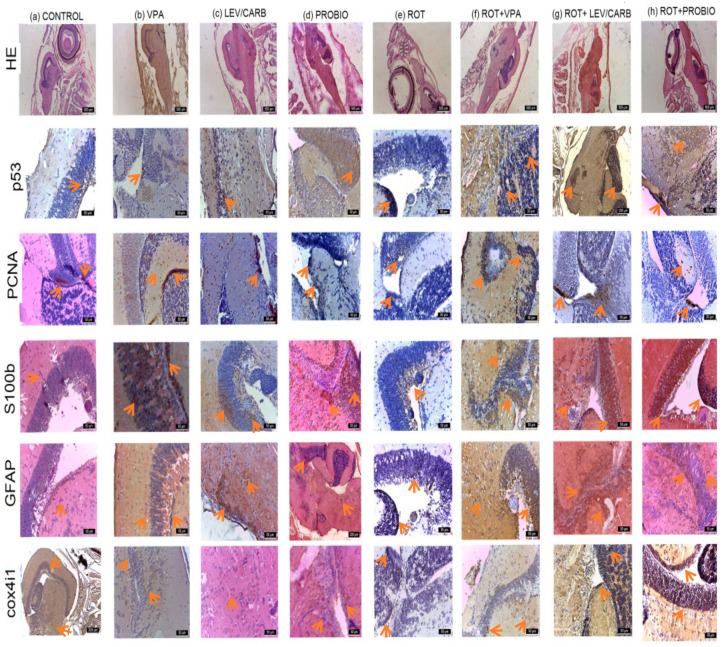
The reactivity of the nervous system in studied *Danio rerio* (*n* = 5) groups of HE, p53, PCNA, S100b, GFAP, and cox4i1.

**Figure 6 antioxidants-11-02040-f006:**
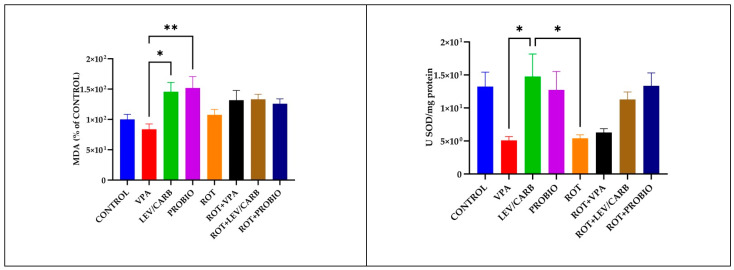
The enzymatic activity of SOD and the level of MDA in the studied *Danio rerio* (*n* = 5) groups (*—*p* < 0.05; **—*p* < 0.005; values expressed as mean with SEM followed Tukey HSD test).

**Figure 7 antioxidants-11-02040-f007:**
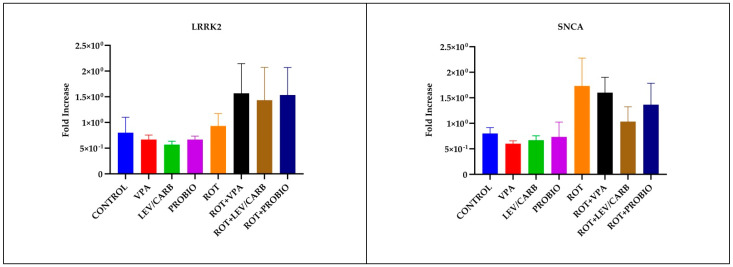
*LRRK2*, *SNCA*, *PARKIN,* and *PINK1* relative expression levels in the brain of studied *Danio rerio* (*n* = 3) groups (*—*p* < 0.05; **—*p* < 0.005; ***—*p* < 0.0005; values expressed as mean with SEM followed by Tukey HSD).

**Figure 8 antioxidants-11-02040-f008:**
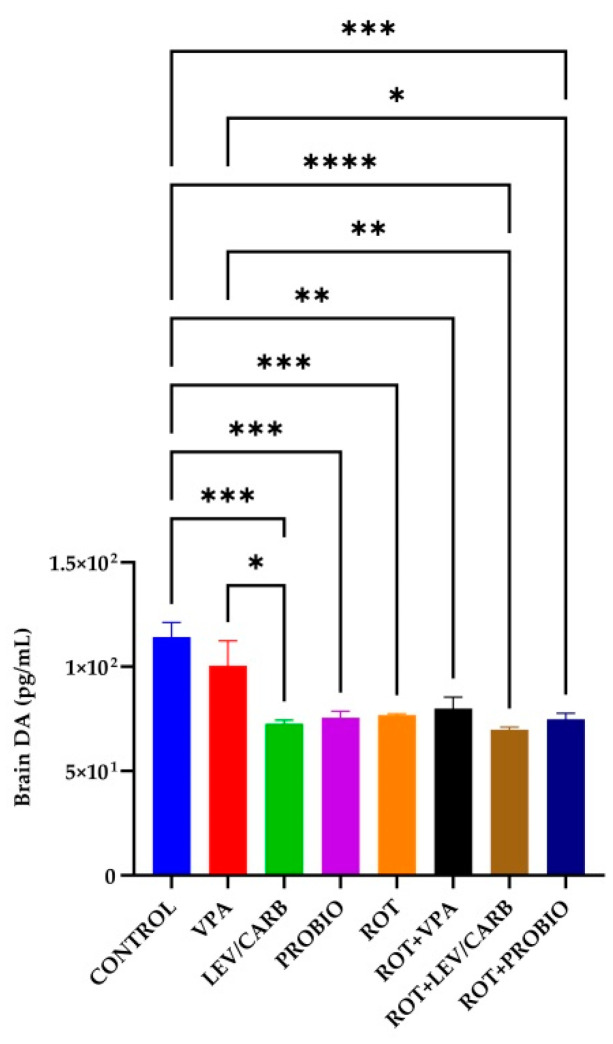
Brain DA level in the studied *Danio rerio* (*n* = 5) groups (*—*p* < 0.05; **—*p* < 0.005; ***—*p* < 0.0005; ****—*p* < 0.0001; values expressed as mean with SEM followed by Tukey HSD).

## Data Availability

The datasets used and analyzed during the current study are available from the corresponding author on reasonable request.

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
