# Peer review of "Assessing the Neurotoxicity of a Sub-Optimal Dose of Rotenone in Zebrafish (Danio rerio) and the Possible Neuroactive Potential of Valproic Acid, Combination of Levodopa and Carbidopa, and Lactic Acid Bacteria Strains"

_antioxidants, 2022, doi:10.3390/antiox11102040_

Round 1
Reviewer 1 Report
The manuscript entitled “Assessing how Valproic acid, Anti-parkinsonian Drugs, and Lactic Acid Bacteria strains might act as dopaminergic agonists through modulation of the gut-brain axis in a zebrafish (Danio rerio)-model of Parkinson’s disease chronically exposed to rotenone” mainly addresses the beneficial neuroprotective effects of PROBIO (Lactic Acid Bacteria strains) in rotenone-triggered Parkinson’s like features in zebrafish. At the molecular levels, the authors investigated oxidative stress markers and neurogenesis.
Comments:
1) The statistical comparisons are not proper. In fact, for each parameter, the readers are looking for statistical significance among the 8 experimental groups on the same day (for example D-32) but not the difference between days for the same group. A good example already done by the authors is figure 6 for the quantification of MDA and SOD. What is really important for readers is whether a statistical significance exists between ROT and control; this indicates the Parkinson’s model exerted a significant change. Equally important, the significant changes between ROT gp vs ROT+VPA, ROT+LEV/VARB, or ROT+PROBIO are important in indicating that these treatments exerted a significant attenuation of oxidative stress, for example, in the brain of zebrafish. Thus, the authors are advised to rewrite the entire results section.
2) The results in figure 6 are problematic in the sense that there is a non-significant change in MDA levels between ROT group vs control. This means that rotenone does not affect the brain’s MDA levels which is against the well-documented literature. More importantly, there are non-significant changes between ROT gp vs ROT+VPA, ROT+LEV/VARB, or ROT+PROBIO. This clearly indicates that VPA, LEV/VARB, and PROBIO do not attenuate brain oxidative stress. Please, clarify this point in the discussion section.
3) The same problematic data exists in figure 6 for the quantification of SOD where there is a non-significant change between ROT group vs control. This means that rotenone does not affect the brain SOD levels which is against the well-documented literature. More importantly, there are non-significant changes between ROT gp vs ROT+VPA, ROT+LEV/VARB, or ROT+PROBIO. This clearly indicates that VPA, LEV/VARB, and PROBIO do not attenuate brain oxidative stress. Please, clarify this point in the discussion section.
4) In figure 6, SOD quantification, the authors are advised to double-check the statistics since it is clear by naked eye that ROT+LEV/VARB, and ROT+PROBIO are higher than ROT gp, suggesting a possible significant difference.
5) The results in figure 8 are problematic. This is clear from the fact that there is non-significant changes in brain dopamine levels between ROT gp vs ROT+VPA, ROT+LEV/VARB, or ROT+PROBIO. This clearly indicates that VPA, LEV/VARB, and PROBIO do not affect brain dopamine levels.
6) In the results section, to avoid confusion among readers, the authors are advised to divide the results section into sections and to clearly label each section.
7) how would data in figure 8 reveal that the PROBIO act as dopamine agonist? First, the figure only investigates the level of dopamine in brain of zebrafish, so how would that reflect the binding affinity of dopamine to its receptor to prove it acts as an agonist?
8) p53 alone is not a specific marker for apoptosis. Authors should have investigated caspase-3 activity or cleaved caspase 3 protein levels.
9) The title of the current study is not confusing and not focused, for the following reasons:
A) The current work only investigates the level of dopamine in the brain of zebrafish, so how would that reflect the binding affinity of dopamine to its receptor to prove it acts as an agonist?
B) Please, remove the gut-brain axis from the title since the present work did not measure of the metabolites produced by these strains.
C) The current study only investigated LEV/VARB combination from the anti-Parkinson drugs, so, it would be inaccurate to keep the whole class of “Anti-parkinsonian Drugs”.
D) The current title should focus on the effect of PROBIO (Lactic Acid Bacteria strains).
10) To make it clearer for readers in Figure 5, quantification of the protein expression in the 8 experimental groups for each target protein is advised. Without quantification of target proteins PCNA, p53, …etc, it is impossible to judge that there is an increase or decrease in the protein expression of any target protein.
11) in line 290, the authors describe that ROT triggers reduction in apoptosis as indicated by decreased p53 levels. This is against the consensus of literature that rotenone triggers neuronal apoptosis. Please, address this controversy in the discussion section.
12) In line 39, the authors describe rotenone as “dopamine antagonist”. This is not accurate since the literature describes that rotenone destroys dopaminergic neurons and induces parkinsonian features. Please, correct this issue in the entire manuscript.
13) In the experimental protocol section, how did the authors decide on the dose of rotenone (2.5 μg/L) in zebrafish? Authors are advised to address this point and add the answers to the comment in section 2.3. Please also provide proper citations for selecting this dose.
14) Likewise, how did the authors decide on the dose of valproic acid, LEV/CARB and PROBIO? Authors are advised to address this point and add the answers to the comment in section 2.3. Please also provide proper citations for selecting this dose.
15) In the introduction section, the authors are advised to clarify for readers how PROBIO affects the nervous tissue for zebrafish despite not crossing the blood-brain barrier.
16) In the statistical analysis section, the authors are advised to clarify for the readers what was the minimum P value. Authors are advised to address this point and add the answers to the comment in the material and methods section.
17) In the PCR: The authors are advised to add the gene accession number and amplicon size for all target genes. Please, add these data in the material and methods section
18) The author should mention the amount of RNA used to synthesize cDNA. Please, add these data in the material and methods section.
19) The qRT-PCR is missing biological (how many samples were used per experimental group) and technical repeat information (whether each sample was repeated during the assay). Moreover, did the authors perform an RT negative control to ensure no DNA contamination in the RNA extraction? Please, add these data in the material and methods section.
20) The investigation of PINK1, PARKIN, LRRK2, alpha-SNCA genes was performed only at the level of mRNA expression. Have the authors considered that the gene expression assays of these genes using RT-PCR may not be adequate for quantifying the target signals since the mRNA expression may not necessarily reflect the corresponding protein levels due to the post-translational modifications? In fact, detecting protein signals using ELISA or Western blotting is expected to give more reliable data than gene expression assays.
Author Response
Dear Reviewer #1,
We would like to thank you very much for the positive feedback, interest, and time spent reviewing our manuscript. All the asked data can be found in the revised version of our manuscript per your instructions. Per your instructions, we made the respective changes that can be found below:
The manuscript entitled “Assessing how Valproic acid, Anti-parkinsonian Drugs, and Lactic Acid Bacteria strains might act as dopaminergic agonists through modulation of the gut-brain axis in a zebrafish (Danio rerio)-model of Parkinson’s disease chronically exposed to rotenone” mainly addresses the beneficial neuroprotective effects of PROBIO (Lactic Acid Bacteria strains) in rotenone-triggered Parkinson’s like features in zebrafish. At the molecular levels, the authors investigated oxidative stress markers and neurogenesis.
Comment from the Reviewer:
1) The statistical comparisons are not proper. In fact, for each parameter, the readers are looking for statistical significance among the 8 experimental groups on the same day (for example D-32) but not the difference between days for the same group. A good example already done by the authors is figure 6 for the quantification of MDA and SOD. What is really important for readers is whether a statistical significance exists between ROT and control; this indicates the Parkinson’s model exerted a significant change. Equally important, the significant changes between ROT gp vs ROT+VPA, ROT+LEV/VARB, or ROT+PROBIO are important in indicating that these treatments exerted a significant attenuation of oxidative stress, for example, in the brain of zebrafish. Thus, the authors are advised to rewrite the entire results section.
Response: Dear Reviewer, we agree that every reader looks for statistical significance between the groups and not the difference within one group on different days. However, as our team has repeatedly demonstrated [31,37,41–46], this approach might offer a conclusive view regarding the spatial conformations in the analyzed groups based on the compound administered and phenotype expected and exhibited by comparing the pretreatment with the treatment period (baseline behavior vs new behavior following exposure to a compound). As we mentioned in the revised version, in most of the existing literature, the authors assessed the locomotor activity by simple observation [48], the locomotor performances being performed once a week (mean) [33,35] or not specified [34,36], nor regarding ROT administration or substance renewal that could be another variable that may influence data and analyses.
Comments from the Reviewer:
2) The results in figure 6 are problematic in the sense that there is a non-significant change in MDA levels between ROT group vs control. This means that rotenone does not affect the brain’s MDA levels which is against the well-documented literature. More importantly, there are non-significant changes between ROT gp vs ROT+VPA, ROT+LEV/VARB, or ROT+PROBIO. This clearly indicates that VPA, LEV/VARB, and PROBIO do not attenuate brain oxidative stress. Please, clarify this point in the discussion section.
3) The same problematic data exists in figure 6 for the quantification of SOD where there is a non-significant change between ROT group vs control. This means that rotenone does not affect the brain SOD levels which is against the well-documented literature. More importantly, there are non-significant changes between ROT gp vs ROT+VPA, ROT+LEV/VARB, or ROT+PROBIO. This clearly indicates that VPA, LEV/VARB, and PROBIO do not attenuate brain oxidative stress. Please, clarify this point in the discussion section.
4) In figure 6, SOD quantification, the authors are advised to double-check the statistics since it is clear by naked eye that ROT+LEV/VARB, and ROT+PROBIO are higher than ROT gp, suggesting a possible significant difference.
Response: Dear Reviewer, we again agree that our results are not congruent with the existing data since most authors reported significance between the control and the exposed zebrafish in MDA and SOD. “Judging by the non-significant difference between (a) CONTROL and the remaining seven experimental groups, we can argue that both the level of MDA and the enzymatic expression of SOD might be stress-related, considering our previous results where we had significance between the CONTROL and the experimental groups following 2 µg/L ROT administration for 21 days and daily testing. Another possible explanation for the lack of significance is that VPA, LEV/CARB, and PROBIO combined with ROT cancel each other’s effect (hypothesis that should be further tested and validated). Thus, it is hard to establish if the administered compound alone or mixture causes an elevation or downregulation in oxidative biomarkers or is the result of prolonged exposure to stress due to daily behavioral testing. In either of these scenarios, the chances of misleading conclusions are high, which is why additional studies are mandatory to delimitate these aspects.” Moreover, we have double-check the statistics for SOD and indeed are multiple other cases when we have a p < 0.5, but not p < 0.05 as we set the baseline.
Comment from the Reviewer:
5) The results in figure 8 are problematic. This is clear from the fact that there is non-significant changes in brain dopamine levels between ROT gp vs ROT+VPA, ROT+LEV/VARB, or ROT+PROBIO. This clearly indicates that VPA, LEV/VARB, and PROBIO do not affect brain dopamine levels.
Response: Dear Reviewer, the expectations are fulfilled if we refer to the difference in dopamine levels between the healthy (CONTROL) group and those exposed to ROT, which further sustains our results. However, these results might be problematic if we refer solely to the ROT-exposed groups when indeed there is no statistically significant difference.
Comment from the Reviewer:
6) In the results section, to avoid confusion among readers, the authors are advised to divide the results section into sections and to clearly label each section.
Response: Dear Reviewer, we divided the Results section into subsections to avoid confusion among readers.
Comment from the Reviewer:
7) how would data in figure 8 reveal that the PROBIO act as dopamine agonist? First, the figure only investigates the level of dopamine in brain of zebrafish, so how would that reflect the binding affinity of dopamine to its receptor to prove it acts as an agonist?
Response: Dear Reviewer, data in figure 8 did not reveal the role of PROBIO as dopamine agonist. Considering the crucial role of probiotics in distinct fields of activity, we hypothesized that lactic acid bacteria strains exogenously administered will influence the commensal bacteria and subsequently, through the lipopolysaccharides synthesized initiate signaling cascades and chemical responses which cross the blood-brain barrier and finally lead to dopamine release. Unfortunately, we did not investigate the gut microflora and its additional mechanisms, but further studies might offer insight into this context.
Comments from the Reviewer:
8) p53 alone is not a specific marker for apoptosis. Authors should have investigated caspase-3 activity or cleaved caspase 3 protein levels.
10) To make it clearer for readers in Figure 5, quantification of the protein expression in the 8 experimental groups for each target protein is advised. Without quantification of target proteins PCNA, p53, …etc, it is impossible to judge that there is an increase or decrease in the protein expression of any target protein.
20) The investigation of PINK1, PARKIN, LRRK2, alpha-SNCA genes was performed only at the level of mRNA expression. Have the authors considered that the gene expression assays of these genes using RT-PCR may not be adequate for quantifying the target signals since the mRNA expression may not necessarily reflect the corresponding protein levels due to the post-translational modifications? In fact, detecting protein signals using ELISA or Western blotting is expected to give more reliable data than gene expression assays.
Response: Based on your suggestions on three distinct points, we added in the manuscript Limitations of the study section surrounding protein quantification using Western blotting.
Comment from the Reviewer:
9) The title of the current study is not confusing and not focused, for the following reasons:
- A) The current work only investigates the level of dopamine in the brain of zebrafish, so how would that reflect the binding affinity of dopamine to its receptor to prove it acts as an agonist?
- B) Please, remove the gut-brain axis from the title since the present work did not measure of the metabolites produced by these strains.
- C) The current study only investigated LEV/VARB combination from the anti-Parkinson drugs, so, it would be inaccurate to keep the whole class of “Anti-parkinsonian Drugs”.
- D) The current title should focus on the effect of PROBIO (Lactic Acid Bacteria strains).
Response: Dear Reviewer, we changed the titled of the manuscript. “Assessing the possible neuroactive potential of Valproic acid, combination of Levodopa and Carbidopa, and of Lactic Acid Bacteria strains in a zebrafish (Danio rerio)-model of Parkinson’s disease chronically exposed to Rotenone”
Comment from the Reviewer:
11) in line 290, the authors describe that ROT triggers reduction in apoptosis as indicated by decreased p53 levels. This is against the consensus of literature that rotenone triggers neuronal apoptosis. Please, address this controversy in the discussion section.
Response: Dear Reviewer, it was a mistake on our part. We wanted to say that ROT engages the expression of p53 and cox4i1 labeling which indicates and emphasizes apoptosis and mitochondrial dysfunction. We made the necessary corrections.
Comment from the Reviewer:
12) In line 39, the authors describe rotenone as “dopamine antagonist”. This is not accurate since the literature describes that rotenone destroys dopaminergic neurons and induces parkinsonian features. Please, correct this issue in the entire manuscript.
Response: Dear Reviewer, we made the necessary corrections throughout the entire manuscript.
Comment from the Reviewer:
13) In the experimental protocol section, how did the authors decide on the dose of rotenone (2.5 μg/L) in zebrafish? Authors are advised to address this point and add the answers to the comment in section 2.3. Please also provide proper citations for selecting this dose.
Response: “As already demonstrated by our group [31] and concomitantly by another team [32], administration of 2 µg/L ROT for 21, up to 28 days triggers a mild locomotor impairment in zebrafish. On the other hand, 5 µg/L ROT [33–36] for 28 [33,35], up to 30 days [34,36] exposure led to visible locomotor dysfunctionalities. Due to excessive mortality approximately ten days since inception that occurred in a preliminary study conducted by us following the administration of 5 µg/L ROT (data not shown), we decided to halve the dose.”
Comment from the Reviewer:
14) Likewise, how did the authors decide on the dose of valproic acid, LEV/CARB and PROBIO? Authors are advised to address this point and add the answers to the comment in section 2.3. Please also provide proper citations for selecting this dose.
Response: “In the same preliminary experiment we also tested four concentrations of VPA (0.5 mg/mL, 2 mg/mL, 5 mg/mL, and 10 mg/mL). In zebrafish receiving 5 mg/mL and 10 mg/mL, we noted high mortality in the first 6-12 hours post-administration, while those exposed to 2 mg/mL were immobile upon touching (data not shown). In this way, we concluded that 0.5 mg/mL might be the optimum dose since a study in which we implied this approach was already published [37].
“In what concerns the associated dose of LEV/CARB, we used as informatic support the study of Idalencio et al. [24] where the authors emphasize the role of LEV/CARB in stress response in zebrafish specific to stress in contrast with non-stressed fish suggesting that DA was related to the balance between high and low cortisol levels and that norepinephrine (NE) decrease this response.”
“There are no restrictions regarding PROBIO dose and ratio of bacteria but rather personalized depending on the study design and product’ strains and colony forming units (CFU) than sex and species. For example, Valcarce et al. [30,38] administered a 1:1 ratio of Lactobacillus rhamnosus CECT8361 and Bifidobacterium longum CECT7347 in both Danio rerio and human patients to assess how short- and long-term administration of these two PROBIO strains improve sperm quality. In another study, Valcarce et al. [39] used the same combination to alter swimming patterns and speed in zebrafish by down-regulating anxiety-related behavior. In one study of ours, Bifidobacterium longum BB536 and Lactobacillus rhamnosus HN001 were administered to 2 µg/L ROT- exposed zebrafish to investigate their possible role in sociability and locomotion [31].”
Comment from the Reviewer:
15) In the introduction section, the authors are advised to clarify for readers how PROBIO affects the nervous tissue for zebrafish despite not crossing the blood-brain barrier.
Response: “Compared with the aforementioned, PROBIO cannot cross the BBB, but the responses initiated influence the brain’s reactivity through the intracellular cascades emitted. According to the evidence in the field, exogenous supplementation with PROBIO enhance the host’s gut microflora integrity by preventing dysbiosis. In this context, microorganisms that reside within the gut negate the pathogens’ endotoxins, which otherwise cross the BBB and trigger neuroinflammation. Thus, a regime might impact the host composition through competition, antagonism, and cross-feeding [26,27].”
Comment from the Reviewer:
16) In the statistical analysis section, the authors are advised to clarify for the readers what was the minimum P value. Authors are advised to address this point and add the answers to the comment in the material and methods section.
Response: Dear Reviewer, p < 0.05 was considered statistically significant.
Comment from the Reviewer:
17) In the PCR: The authors are advised to add the gene accession number and amplicon size for all target genes. Please, add these data in the material and methods section
Response: Dear Reviewer, we added the requested information. PINK1 (NM_001008628.1) (2088 bp ), f: 5’-GGCAATGAAGATGATGTGGAAC-3’, r: 5’-TTGTGGGCATGAAGGAACTAAC-3’, PARKIN (NM_001017635.1) (1465 bp), f: 5’-GAGGAGTTTCACGAGGGTCC-3’, r: 5’-TGAGTGGTTTTGGTGATGGTC-3’, LRRK2 (NM_001201456.2) (9170 bp), f: 5’-ACTCGGATTAAGTTCCACCAGA-3’, r: 5’-CAGTGAGGGTTGATGGTCTGTA-3’, alpha-SNCA (NM_001017567.2) (1294 bp), f: 5’-ATGCACTGAAGAAGGGATTCTC-3’, r: 5’-AGATTTGCCTGGTCAGTTGTTT-3’, and ACTIN (NM_181601.5) (1843 bp), f: 5’-GGCATCACACCTTCTACAATGA-3, r: 5’-TACGACCAGAAGCGTACAGAGA-3’.
Comment from the Reviewer:
18) The author should mention the amount of RNA used to synthesize cDNA. Please, add these data in the material and methods section.
Response: Dear Reviewer, an amount of 4 µl of RNA was used to synthesize the cDNA.
Comment from the Reviewer:
19) The qRT-PCR is missing biological (how many samples were used per experimental group) and technical repeat information (whether each sample was repeated during the assay). Moreover, did the authors perform an RT negative control to ensure no DNA contamination in the RNA extraction? Please, add these data in the material and methods section.
Response: Dear Reviwer, each analysis being conducted in duplicate using n = 5 individuals, except RT-PCR where we used only n = 3 individuals. We implied RT negative controls for each targeted gene(s) to ensure no DNA contamination.
Kind regards and all the best,
Ovidiu-Dumitru Ilie
Reviewer 2 Report
The manuscript "aims to offer systematic insights by assessing VPA, 96 LEV/CARB, and PROBIO effects in a zebrafish (Danio rerio) model of PD chronically 97 exposed to ROT for 32 days." Eight groups of zebrafish were analyzed for behavioral responses to rotenone as a DA antagonist and the impact of valproic acid, antiparkinsonian drugs and live active lactic acid bacteria strains examined. The results are clearly presented, though the combinations are sometimes complex to decipher simply due to the number of combined treatments considered. The authors conclusions are supported by the data and represent an extension of the field.
No major problems appear with the manuscript.
A minor confusion is the difference in initial movement values under each condition (relative to each other). Either I missed the explanation of how the movement was measured relative to treatments and the 32 day sampling times, or it was not presented clearly in the methods or data presentation. That is, is the initial value a control prior to treatment for 32 days? If not, what is the reason for differences in initial values?
Otherwise the methods are exemplary in detail.
The use of histochemistry could use a bit more highlighting of specific changes, as there is a great deal of imaging without highlighted regions illustrating what the reader should look for in each image relative to the biology. Please add the descriptive (increased brown staining) or add an arrows to specifically teach the reader where "(PCNA) marks two small areas of neuronal stem cells (NSC) and neuroblasts, respectively" as stated in lines 260-261. We can be convinced, but the color patterns change enough so as to make interpretation a bit more challenging than needed.
Author Response
Dear Reviewer #2,
We would like to thank you very much for the positive feedback, interest, and time spent reviewing our manuscript. All the asked data can be found in the revised version of our manuscript per your instructions. Per your instructions, we made the respective changes that can be found below:
Comment from the Reviewer: The manuscript "aims to offer systematic insights by assessing VPA, 96 LEV/CARB, and PROBIO effects in a zebrafish (Danio rerio) model of PD chronically 97 exposed to ROT for 32 days." Eight groups of zebrafish were analyzed for behavioral responses to rotenone as a DA antagonist and the impact of valproic acid, antiparkinsonian drugs and live active lactic acid bacteria strains examined. The results are clearly presented, though the combinations are sometimes complex to decipher simply due to the number of combined treatments considered. The authors conclusions are supported by the data and represent an extension of the field.
Response: Dear Reviewer, thank you very much for your time, feedback, interest and kind words toward our manuscript.
Comment from the Reviewer: No major problems appear with the manuscript.
Response: Thank you.
Comment from the Reviewer: A minor confusion is the difference in initial movement values under each condition (relative to each other). Either I missed the explanation of how the movement was measured relative to treatments and the 32 day sampling times, or it was not presented clearly in the methods or data presentation. That is, is the initial value a control prior to treatment for 32 days? If not, what is the reason for differences in initial values?
Response: Dear Reviewer, we followed and analyzed the possible spatial conformations of the behavior in perspective (D1......D32) by comparing the initial behavior, as pretreatment to the treatment period within one group. This method was applied for each experimental group. Based on our experience, this approach offers a conclusive view [31,37,41–46] depending on the compound administered and phenotype expected and exhibited. As we mentioned in the revised version, in most of the existing literature, the authors assessed the locomotor activity by simple observation [48], the locomotor performances being performed once a week (mean) [33,35] or not specified [34,36], nor regarding ROT administration or substance renewal that could be another variable that may influence data and analyses.
Comment from the Reviewer: Otherwise the methods are exemplary in detail.
Response: Thank you.
Comment from the Reviewer: The use of histochemistry could use a bit more highlighting of specific changes, as there is a great deal of imaging without highlighted regions illustrating what the reader should look for in each image relative to the biology. Please add the descriptive (increased brown staining) or add an arrows to specifically teach the reader where "(PCNA) marks two small areas of neuronal stem cells (NSC) and neuroblasts, respectively" as stated in lines 260-261. We can be convinced, but the color patterns change enough so as to make interpretation a bit more challenging than needed.
Response: Dear Reviewer, we added arrows in accordance with the text to ease the reader's efforts in this manuscript.
Kind regards and all the best,
Ovidiu-Dumitru Ilie
Round 2
Reviewer 1 Report
The authors did not properly address the comments of the reviewer. This is for the following reasons:
I) The authors have not addressed the previous comment # 1. Based on the way the authors are describing data and making statistical comparisons, the current conclusions are not valid.
“Comment # 1:
1) The statistical comparisons are not proper. In fact, for each parameter, the readers are looking for statistical significance among the 8 experimental groups on the same day (for example D-32) but not the difference between days for the same group. A good example already done by the authors is figure 6 for the quantification of MDA and SOD. What is really important for readers is whether a statistical significance exists between ROT and control; this indicates the Parkinson’s model exerted a significant change. Equally important, the significant changes between ROT gp vs ROT+VPA, ROT+LEV/VARB, or ROT+PROBIO are important in indicating that these treatments exerted a significant attenuation of oxidative stress, for example, in the brain of zebrafish. Thus, the authors are advised to rewrite the entire results section”.
II) Several parameters that were measured by the authors revealed non-significant differences which is completely against the consensus of many published studies. This is clear from the previous comments # 2, 3, and 4.
“2) The results in figure 6 are problematic in the sense that there is a non-significant change in MDA levels between ROT group vs control. This means that rotenone does not affect the brain’s MDA levels which is against the well-documented literature. More importantly, there are non-significant changes between ROT gp vs ROT+VPA, ROT+LEV/VARB, or ROT+PROBIO. This clearly indicates that VPA, LEV/VARB, and PROBIO do not attenuate brain oxidative stress. Please, clarify this point in the discussion section.
3) The same problematic data exists in figure 6 for the quantification of SOD where there is a non-significant change between ROT group vs control. This means that rotenone does not affect the brain SOD levels which is against the well-documented literature. More importantly, there are non-significant changes between ROT gp vs ROT+VPA, ROT+LEV/VARB, or ROT+PROBIO. This clearly indicates that VPA, LEV/VARB, and PROBIO do not attenuate brain oxidative stress. Please, clarify this point in the discussion section.
4) In figure 6, SOD quantification, the authors are advised to double-check the statistics since it is clear by naked eye that ROT+LEV/VARB, and ROT+PROBIO are higher than ROT gp, suggesting a possible significant difference”.
III) Lack of selection of the proper markers of apoptosis as described in previous comment # 8.
Comment # 8:
“8) p53 alone is not a specific marker for apoptosis. Authors should have investigated caspase-3 activity or cleaved caspase 3 protein levels”.
IV) Even when the reviewer asked for quantification of the immunostaining (comment # 10 in the previous review), that point was not addressed.
“Comment # 10:
10) To make it clearer for readers in Figure 5, quantification of the protein expression in the 8 experimental groups for each target protein is advised. Without quantification of target proteins PCNA, p53, …etc, it is impossible to judge that there is an increase or decrease in the protein expression of any target protein”.
V) Controversial observation that rotenone lowers apoptosis! This is again against the scientific main consensus in the literature that rotenone increases apoptosis (comment # 11 in the previous review).
“Comment # 11:
11) in line 290, the authors describe that ROT triggers a reduction in apoptosis as indicated by decreased p53 levels. This is against the consensus of literature that rotenone triggers neuronal apoptosis. Please, address this controversy in the discussion section”.
For the previous reasons, the reviewer is not satisfied with the authors’ reply and revision of the manuscript. Thus, the manuscript is not ready as it currently stands for publication.
Author Response
Dear Reviewer #1,
Please see the attached document.

Round 3
Reviewer 1 Report
The revised version of the manuscript has been modified to reflect the obtained data and conclusions better.